# Performance of a Full-Scale Upstream MAPS-Based Verification Device for Radiotherapy

**DOI:** 10.3390/s23041799

**Published:** 2023-02-06

**Authors:** Jaap Velthuis, Yutong Li, Jordan Pritchard, Chiara De Sio, Lana Beck, Richard Hugtenburg

**Affiliations:** 1School of Physics, University of Bristol, Bristol BS7 1TL, UK; 2Swansea University Medical School, Faculty of Medicine, Health and Life Science, Swansea University, Swansea SA2 8PP, UK; 3School of Nuclear Science and Technology, University of South China, Hengyang 421001, China; 4Department of Medical Physics and Clinical Engineering, Swansea Bay University Health Board, Swansea SA2 4QA, UK

**Keywords:** X-ray detectors, solid-state detectors, radiation-hard detectors, image processing, data processing methods, image reconstruction in medical imaging, radiotherapy concepts, radiotherapy verification, radiotherapy monitoring, detector alignment and calibration, Multi Leaf Collimator (MLC), Monolithic Active Pixel Sensors (MAPS)

## Abstract

Intensity-modulated radiotherapy is a widely used technique for accurately targeting cancerous tumours in difficult locations using dynamically shaped beams. This is ideally accompanied by real-time independent verification. Monolithic active pixel sensors are a viable candidate for providing upstream beam monitoring during treatment. We have already demonstrated that a Monolithic Active Pixel Sensor (MAPS)-based system can fulfill all clinical requirements except for the minimum required size. Here, we report the performance of a large-scale demonstrator system consisting of a matrix of 2 × 2 sensors, which is large enough to cover almost all radiotherapy treatment fields when affixed to the shadow tray of the LINAC head. When building a matrix structure, a small dead area is inevitable. Here, we report that with a newly developed position algorithm, leaf positions can be reconstructed over the entire range with a position resolution of below ∼200 μm in the centre of the sensor, which worsens to just below 300 μm in the middle of the gap between two sensors. A leaf position resolution below 300 μm results in a dose error below 2%, which is good enough for clinical deployment.

## 1. Introduction

External Beam Radiotherapy (EBRT) is a cancer therapy in which ionising radiation is used to target malignant tissue. EBRT plays a key role in the treatment of cancer. Over 50% of all cancer patients are prescribed EBRT and it contributes towards 40% of all cured cases [1]. In X-ray radiotherapy, a linear accelerator (LINAC) is used to generate an X-ray beam. The X-ray beam interacts throughout the body and thus damages both cancerous and healthy tissue. To minimise the dose to the healthy tissue, the LINAC is moved around the patient such that the beam enters the body at different locations. The beam is shaped according to the tumour profile from each direction using a Multi-Leaf Collmimator (MLC). An MLC consists of two opposing rows of 40–120 uniform tungsten leaves, normally of 5 mm width at the isocentre, which move independently to create complex treatment beam geometries.

To successfully monitor intensity-modulated radiotherapy treatments, two key aspects need verification: the positions of the MLC leaves and dose distribution of the treatment beam. This process of verification is collectively known as in vivo dosimetry with current avenues exploring monitoring devices that can be placed downstream (transit dosimetry) and upstream (transmission dosimetry) of the patient. Transmission dosimetry systems, which are normally attached to the exit window of the LINAC head, are discussed in detail in a previous publication [2], include several commercial offerings. Transit dosimetry systems typically employ the electronic portal imaging device (EPID) that is present on most modern clinical LINAC. A detailed discussion on EPIDs can be found in [3]. In both cases, the radiotherapy treatment is verified as the LINAC gantry rotates around the treatment couch.

The use of transmission detectors in the context of in vivo dosimetry, and in particular pixelated silicon thin-detector technologies, is still quite restricted. A recent publication provides an update on the use of a commercial silicon diode transmission dosimetry system [4] with another offering a solution in a silicon strip detector array [5]. The systems generate significant attenuation of the therapy beam, which would require correction of the treatment plan if used at the time of treatment. A multicentre publication [6] analysed the requirements for the future development of in vivo dosimetry, which it defines as a radiation measurement that is acquired while the patient is being treated. The authors argued that the definition implied that the measurement must be able to capture errors due to patient positioning errors and patient anatomy changes, excluding transmission dosimetry devices, despite the fact that patient positioning systems, including cone beam CT and light-based surface-guidance, are in routine use. A recent UK-based survey of online treatment monitoring solutions for IMRT/VMAT [7] concluded that EPID-based solutions remained the front-runner, while transmission methods were in use in 6% of the centres surveyed. Interestingly, 15% of the respondents without an active EPID dosimetry programme did not consider it sufficiently useful to implement, while 15% of those centres with an active EPID dosimetry programmes did not regard EPID dosimetry as a gold standard for in vivo dosimetry.

The argument that EPID dosimetry solutions collect the dose that has passed through the patient and are therefore sensitive to patient positioning is faulty. Indeed, the sensitivity of the EPID dosimetry method has been called into question, given the long-term stability of EPID systems [8] and the significant departures from tissue equivalence for dosimetry in the amorphous silicon panel [9]. EPIDs are not specifically designed for patient dosimetry, and with better methods for managing patient position, they may be at risk of becoming obsolete. Several authors have demonstrated redesigned downstream dosimetry systems based on silicon with attempts to address such issues [10,11]. A multicentre study of commercial EPID dosimetry systems [12] found out-of-tolerance errors of up to 25%, which was due predominantly to patient anatomical changes and positioning errors; however, it concluded that EPID dosimetry was effective in intercepting important errors. One further study examined the confounding effects of the uncertainties associated with patient anatomical changes [13], commenting that there is currently limited ability to identify and to quantify disagreements caused by position errors.

An upstream monitoring device must be radiation hard enough to operate long enough in a clinical setting, have an attenuation less than 1% to avoid beam hardening, a MLC leaf precision better than 300 μm to keep total dose errors below 2% [14] and be large enough to monitor full treatment fields, which at most measure 40 × 40 cm^2^ at the isocentre. As demonstrated before, a Monolithic Active Pixel Sensor (MAPS)-based device can fulfill all requirements, except the size; see for example [2,15,16,17,18,19]. The concept was patented in [20,21]. Dosimetry with the MAPS was shown in [22,23] and patented in [24].

To address the size challenge, a sensor matrix structure was built that has a large enough area. Here, its performance is reported and shown to be sufficient for clinical deployment.

## 2. A Full-Scale Demonstrator Device

The Lassena MAPS [25] measures 12 × 14 cm^2^ and is three-side buttable. A 2 × 2 configuration of Lassenas can monitor full treatment fields and is hence clinically deployable. Such a system was built in a partnership between the University of Bristol and Nordson, which is known as the demonstrator. A frame was constructed to house the sensor array. To protect the sensors and provide additional structural support to prevent sensors flexing or shifting, a 9.5 mm thick layer of styrofoam with bonded 50 μm aluminium foil on one side and 50 μm kapton foil on the face in contact with the sensors was used. The foil faces prevent the foam from bending too much but do not, significantly, add to the weight or attenuation of the panels. The metal frame was designed such that the device could be attached to the LINAC head in place of the shadow tray, i.e., at the exit window of the LINAC head, as seen in Figure 1, which is the final intended position for the proposed MAPS device. Side mounts allowed for the readout electronics to be securely fixed outside of the treatment beam. The device can be securely attached to the LINAC head such that it would remain in place when the LINAC gantry rotated during treatment.

## 3. Performance of the Full-Scale Demonstrator

As mentioned before, for clinical deployment, it is important that the attenuation of the entire system is less than 1% and that the leaf position resolution at the isocentre is below 300 μm. Here, we demonstrate that the device attenuation and the leaf position resolution are sufficient for clinical deployment, even in the inevitable dead area between sensors.

### 3.1. Device Attenuation

Prior to final assembly, the attenuation of the demonstrator was investigated in a radiotherapy treatment suite at the Royal Cornwall Hospital. The device was attached to the head of a Varian LINAC with the additional support materials, and a 200 μm silicon wafer was used in place of the Lassena sensor array. A Farmer ionisation chamber was placed in the centre of a tissue-equivalent phantom and positioned at the isocentre on the treatment couch.

A 5 × 5 cm^2^ field was collimated and aligned with the ionisation chamber. Fields were delivered with and without the demonstrator present. The dose delivered to the phantom was derived using the ionisation chamber and the attenuation of the treatment beam due to the demonstrator determined. This was performed for several beam modalities: 6 MV, 10 MV, 6 FFF and 10 FFF treatment beams. Flattening filter-free (FFF) beams are made by removing the flattening filter. This increases the intensity per pulse of the beam. Attenuations of 0.70 ± 0.05%, 0.59 ± 0.03%, 1.08 ± 0.02%, and 0.78 ± 0.03% were obtained, respectively. The variation in attenuation is expected due to the differences in beam energy and the resulting beam-hardening effects. The Lassena sensors can be easily back-thinned to 200 μm or less, ensuring the demonstrator in its current form has a clinically insignificant attenuation of ⩽1% for a range of beam energies and treatment types [26].

### 3.2. Leaf Edge Position Resolution with Leaves Perpendicular to
the Dead Area

When mounting sensors in a matrix structure, it is inevitable to have a (small) dead area and misalignments in terms of rotations and offsets. This leads to dead areas affecting the leaf position reconstruction when the leaves move perpendicular to the dead area and when the leaves move along the dead area; see Figure 2. To investigate the leaf edge position reconstruction of the array performance, the demonstrator device was placed on the treatment couch of a 160-leaf Elekta Agility LINAC at Swansea Bay University Health Board at ∼60 cm from the photon source. A 20 cm wide treatment field with a beam energy of 6 MV and dose rate of 70 MU/min was selected. All data were taken with the flattening filter in place. MLC leaves of 5 mm width at the isocentre were extended from both banks into the treatment field. Data was recorded at 34 fps. Dark data frames were recorded between each data set. Both Lassenas are aligned to the frame of the MLC leaves by fitting straight lines along the leaf edges. Lassena 1 is rotated counterclockwise by 0.005, and Lassena 2 is rotated clockwise by −0.002 radians, respectively. The dead area between the two active regions was found to be less than 1 mm at the isocentre.

A detailed description of the position reconstruction algorithm can be found in [2]. In summary, first, a pedestal and rolling shutter correction are applied. Next, a bad pixel mask was applied. To reduce event by event pixel signal variation, the data sets were summed in sets of five frames. Reconstruction methods are therefore based on 0.15 s of treatment data. The pixel signal variations are due to the limited number of photons and electrons that interact with a single pixel in a frame, leading to shot noise. Frames are then smoothed using a Gaussian blur with a kernel size of 31 pixels and a sigma of 7. Sobel operators are then applied in the *x* and *y* plane, and its magnitude is calculated. The leaf edges are defined as the location with the highest Sobel output. The standard deviation of the reconstructed position yields the resolution.

#### 3.2.1. The Matching Index Method

However, the Sobel method uses a Gaussian blur with a radius of 31 pixels. This does not work within 31 pixels from either edge, leading to a dead area given by the physical gap between the sensors, the insensitive part of the sensor and 62 pixels. The latter alone results in a dead area of 5.6 mm at the isocentre, which is wider than an MLC leaf. Hence, a new method is required.

Even though there is a small gap between the sensors, the high–low transition between areas covered and not covered by the leaves is not abrupt. The intensity transition is wider than the total gap, as shown in Figure 3. The complete *S*-curves of the MLC leaf edges are clearly visible for extensions where the leaf is not close to the sensor edge such as for example for the 47.5 and 47.1 mm extensions. For extensions close to the edge, only part of the *S*-curve is visible, while for certain extensions, the transitions are spanning both sensors. In Figure 3c, the signal profiles in the two sensors are put side by side. The black dotted line indicates the interface between the sensors. The dead area results in a jump in the signal profiles.

The shape of the *S*-curve is the same for all extensions close to the edge. Hence, the leaf position close to the edge can be obtained by calculating the reconstructed position of a leaf close to the edge but within the Sobel range, such as the 47.5 mm extension in Figure 3 and then for further extensions obtain the shift of the *S*-curve with respect to the reference extension: here, 47.5 mm. This is performed by evaluating $M_{i}$ for each shift (δ) of the reference *S*-curve.
(1)$$M_{i}\left(\delta \right)=\sum_{i=ref.\;pos-75}^{i=ref.\;pos+75}{\left(Ref(i+\delta )-Meas(i)\right)}^{2}$$
where *i* is the pixel index.

A quadratic fit is applied to find the shift distance that minimises $M_{i}$. The position is obtained as the reconstruction position from the reference leaf position plus the shift distance.

For all leaves, the set position was reconstructed 60 times for each extension. Figure 4 shows the reconstructed position resolution for four leaves. The results show that the resolution is between 100 and 200 μm outside of the gap. In the gap, the position resolution increases to ∼300 μm. Since there is a small angle between the sensors, the gap is in slightly different locations for the leaves.

Figure 5 shows the resolution for leaf 1, 5, 8 and 12 using both the Sobel and the matching index methods. Since the matching index method is based on the position reconstructed with the Sobel method, the resolution obtained with the Sobel method is slightly better than the resolution obtained with the matching index method. However, the matching index method can reconstruct the leaf position over the entire leaf set position range. The resolution does become worse in the gap, but it is still below the critical 300 μm to keep the dose error below 2%.

#### 3.2.2. Overall Position Resolution with Leaves Perpendicular to
the Dead Area

From a clinical perspective, it is important to correctly reconstruct the position over the entire sensor area as well as possible. To obtain the best position resolution, the Sobel method was used away from the edge where the matching index method was used. The reconstructed MLC leaf position versus the set position at the isocentre is shown in Figure 6 for leaf 1. The transition between Lassena 2 and Lassena 1 is at a set position of 38.364 mm. A linear fit is made to the data to illustrate that the correspondence between the reconstructed and the set position over both Lassenas returns the correct position. The fit has a gradient of 1.00 ± 0.003 with an intercept of 0.48 ± 0.11, clearly showing that the set positions are reconstructed correctly. The residual as a function of the set position shows that the quality of the position reconstruction is quite homogeneous over the entire range. Fitting the residual distribution with a Gaussian yields a standard deviation and thus resolution of 89 ± 9 μm. This was repeated for all MLC leaves. The performance of all MLC leaves is similar, and resolutions of around 100 μm at the isocentre are obtained. This clearly shows that the our matrix system performs more than good enough for clinical deployment.

### 3.3. Overall Position Resolution with Leaves along
the Dead Area

Since the 2 × 2 matrix has small gaps between all four sensors, there are gaps as well that run along the leaves; i.e., part of the end of the leaf is not observed, see Figure 2 (case B). To test the effect on the leaf position resolution, a similar experiment with the same demonstrator and the same LINAC was undertaken. The leaf positions were reconstructed using the Sobel method for each pixel row or slice. Then, the overall leaf shape is fitted to obtain the final leaf position. Figure 7a shows the leaf edge and fit to determine the leaf position over the sensor gap. The missing region yields a larger variation in the reconstructed position and thus a worse resolution. This is shown in Figure 7b, which shows the resolution as a function of the *x* position of the leaf. The result shows that the leaf resolution approximately doubles to 253 ± 24 μm when the dead area is centrally over the leaf. This increase is well within clinical tolerance of 300 μm, and thus, the system performance is sufficient for clinical deployment.

## 4. Conclusions

We are developing a large-scale, MAPS-based real-time radiotherapy treatment verification device. MAPS can perform the key tasks of a real-time radiotherapy treatment verification device for radiotherapy. For such a device to be clinically deployable, the device must be large enough to monitor full treatment fields, have a leaf position resolution over the entire range below 300 μm, and have an attenuation below 1%. A 2 × 2 sensor matrix structure was built with sufficient area.

Using the same mechanics and a 200 μm thick silicon dummy, it was shown that the device can be made with low enough attenuation. Attenuations of 0.70 ± 0.05%, 0.59 ± 0.03%, 1.08 ± 0.02% and 0.78 ± 0.03% were obtained, respectively, for 6 MV, 10 MV, 6 FFF and 10 FFF treatment beams. The variation in attenuation is expected due to the differences in beam energy and resulting beam-hardening effects. The Lassena sensors can be easily back-thinned to 200 μm or less without loss of performance, ensuring the demonstrator in its current form has a clinically insignificant attenuation of <1% for a range of beam energies and treatment types.

Our previously used Sobel-based algorithm yields leaf position resolutions below ∼150 μm at the isocentre. However, it uses a Gaussian blur with a kernel size of 31 pixels and thus cannot reconstruct leaf positions close to the edge. Using a new leaf position reconstruction algorithm for leaves moving perpendicular to the dead area, leaf positions can be reconstructed over the entire range albeit with a slightly worse position resolution of below ∼200 μm. Even at the most difficult locations, in the middle of the gap between two sensors, the reconstructed position resolution remains below 300 μm. Measuring the overall position resolution for all MLC leaves, resolutions of around 100 μm at the isocentre are obtained. For leaves moving along the dead area, positions can be reconstructed using the Sobel method. Even in the most challenging configuration, when the gap is centrally over the leaf, a leaf resolution of 253 ± 24 μm is obtained.

It was demonstrated that an MAPS-based system can indeed be produced with large enough area, low attenuation and sufficient position resolution. Even in the most challenging situation, the position resolution is better than 300 μm which is the critical value to keep dose errors below 2% and make it clinically deployable.

## Figures and Tables

**Figure 1 sensors-23-01799-f001:**
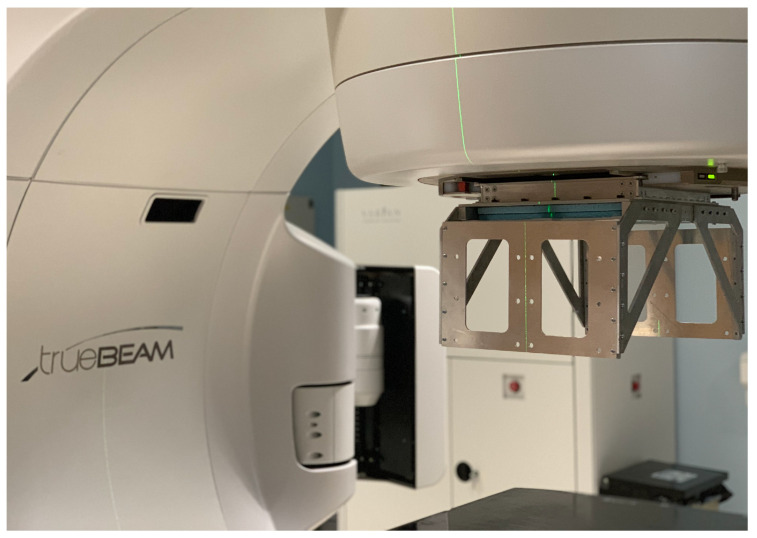
A closeup of the demonstrator device attached to the head of a Varian trueBEAM LINAC. The device is securely attached to the exit window of the LINAC head. The two blue foam layers used to protect the MAPS can be seen. The sensors are in between the two foam layers. The aluminium frame is designed to house the readout electronics (not included here), which is mounted outside of the beam.

**Figure 2 sensors-23-01799-f002:**
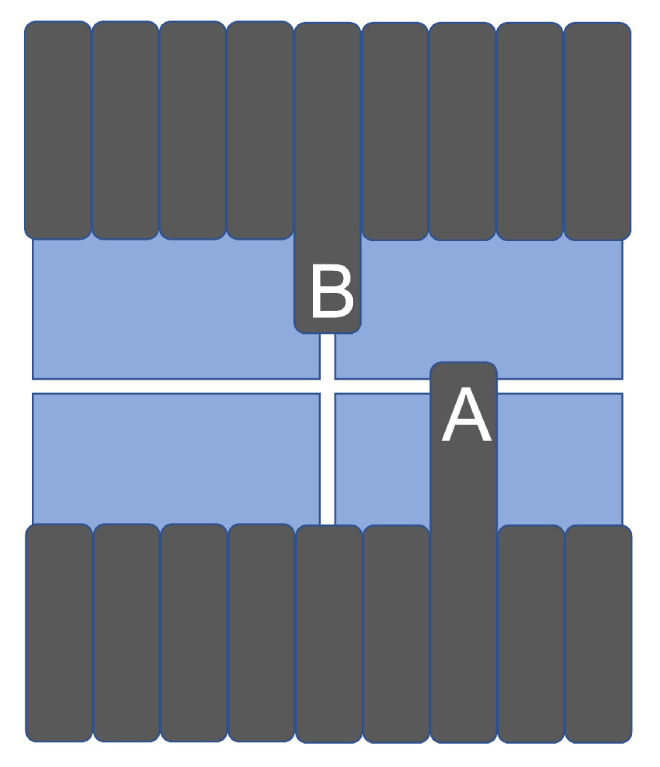
The dead areas lead to leaf position reconstruction issues when the leaves move perpendicular to the dead area (case A) and along the dead area (case B).

**Figure 3 sensors-23-01799-f003:**
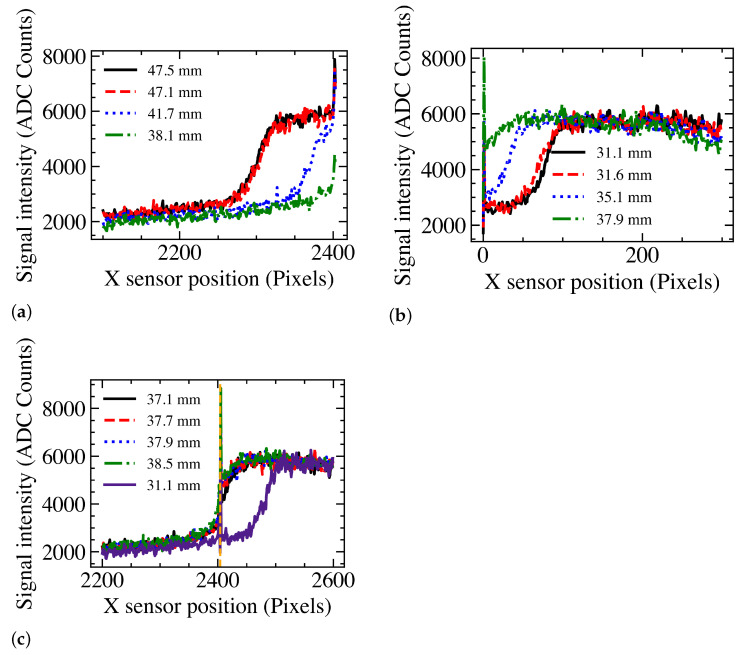
Projection of an MLC leaf edge in one set of 5 averaged frames for a leaf moving from Lassena 2 to Lassena 1 at various leaf extensions. Signal for Lassena 2 in (**a**). Signal for Lassena 1 in (**b**). The legend gives the leaf extensions. (**c**) shows the signal on both Lassenas for several leaf extensions. The black dotted line shows that transition and dead area between the sensors.

**Figure 4 sensors-23-01799-f004:**
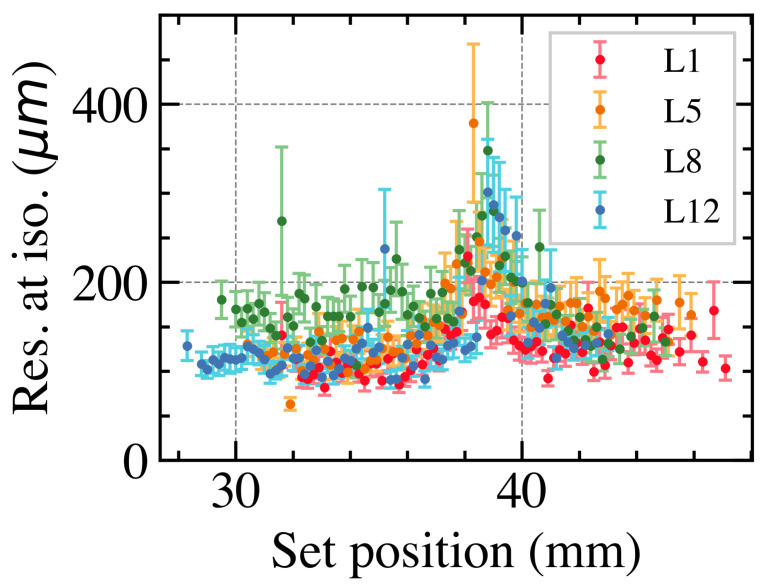
Reconstructed MLC leaf position resolution as a function of the set position for 4 leaves.

**Figure 5 sensors-23-01799-f005:**
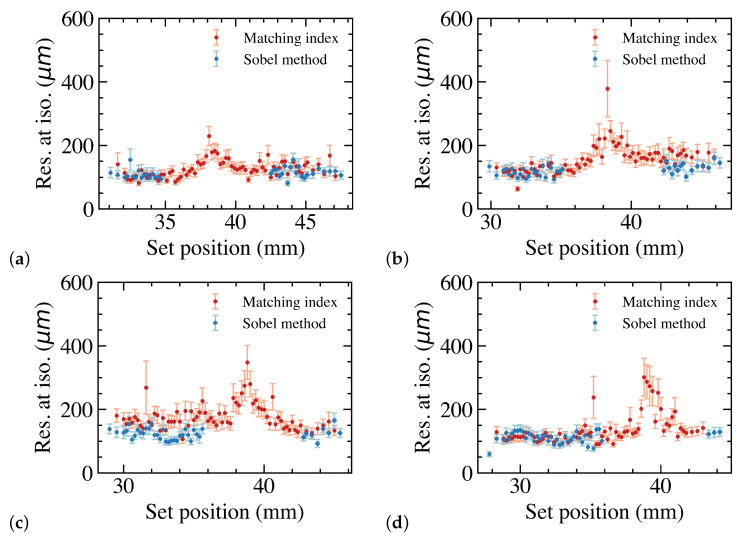
Reconstructed MLC leaf edge position as a function of the set leaf position for leaf 1. The transition between Lassena 2 and Lassena 1 is at a set position of 38.364 mm (**a**). Leaf position residual as a function of the set position for leaf 1. The transition between Lassena 2 and Lassena 1 is at a set position of 38.364 mm (**b**). Residual distribution for leaf 1. A Gaussian fit yields a resolution of 89 ± 9 μm (**c**). Resolution as measured over the entire range as a function of the MLC number (**d**).

**Figure 6 sensors-23-01799-f006:**
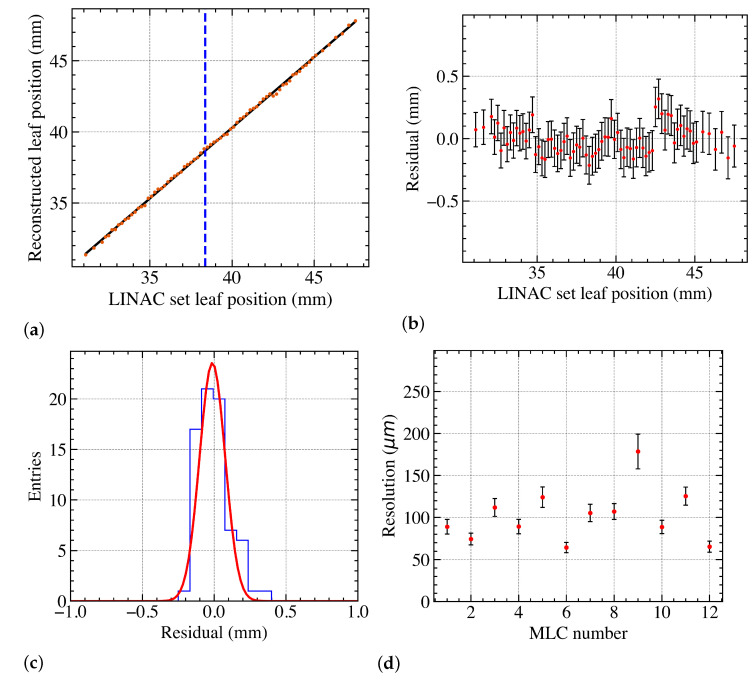
MLC leaf position resolution as a function of the set leaf position for leaf 1 (**a**), leaf 5 (**b**), leaf 8 (**c**) and leaf 12 (**d**) for both the Sobel and matching index methods.

**Figure 7 sensors-23-01799-f007:**
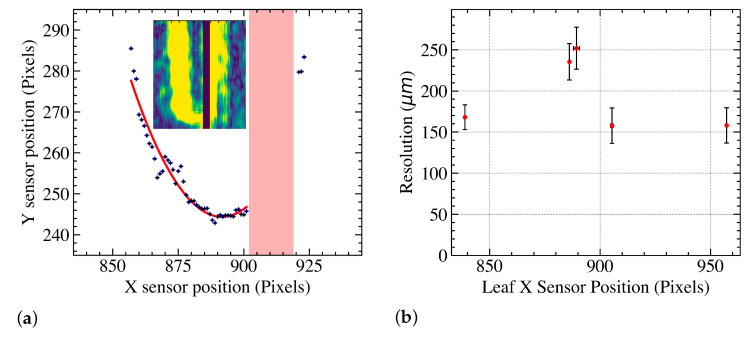
Reconstructed leaf edge over sensor gap and fit to determine leaf position. Sensor gap is overlaid in red. The inset shows the Sobel output around the leaf including the sensor gap (not at the same scale) (**a**). Leaf edge resolution for a leaf sweeping across the sensor gap (**b**). For approximately *x* = 886, 889 and 905 pixels, the gap covers the left, the middle and the right part of the leaf.

## Data Availability

Not applicable.

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
