# Peer review of "Performance of a Full-Scale Upstream MAPS-Based Verification Device for Radiotherapy"

_sensors, 2023, doi:10.3390/s23041799_

Round 1

Reviewer 1 Report

Comments:

This study is an effort to investigate the performance accuracy of a large-scale demonstrator system consisting of a matrix of 2*2 sensors, resulting in a sensor measuring 24*28 cm2 that is highly enough to cover LINAC radiotherapy treatment fields. The proposed idea is interesting and can be applicable for practical cases. The obtained results have high scientific importance degree and can be considered for publication at this journal. In my opinion the following points can further improve the work:

The novelties of the work are not highlighted mainly at Introduction and discussion sections. Please clarify the novelties by adding informative texts. 

The structure of the Introduction section can be revised in order to make better connectivity between the texts. Please also add a last paragraph at introduction section including local discussion on obtained results, conceptually.

At figure 6, why A Gaussian fit is applied to estimate the peak positions. Please explain in the text.

Please add texts about the reason for choosing the implemented registration method.

Please re-generate figure 13 and 16 to improve visualization of the axes explanations.

Please add further references in this work by adding mainly updated works

Author Response

Please see attached our replies.

Reviewer 2 Report

Manuscript Number: sensors-2095543

Full Title: Performance of a full-scale upstream MAPS-based verification device for radiotherapy

Authors presented called "Performance of a full-scale upstream MAPS-based verification device for radiotherapy" in their paper. The authors developed a large-scale, MAPS-based real-time radiotherapy therapy validation device. The main tasks of this system are to monitor MLC leaf positions and treatment beam dose distribution. My reviews and suggestions about their publications are listed ;

·         First of all, Literature research is insufficient. the contribution of the article to the literature is low. The authors should polish the manuscript to improve its writing. The quality of the article should be increased. 

·         Authors should emphasize contribution and novelty in the abstract. Also, Authors should provide more numerical values in the abstract and conclusions sections. State the novelty of your work. The abstract is very similar to the following article.

o   Velthuis, J. J., Beck, L., Hugtenburg, R., Pritchard, J., & De Sio, C. (2020, October). Real-time, upstream, radiotherapy verification using a Monolithic Active Pixel Sensor System. In Journal of Physics: Conference Series (Vol. 1662, No. 1, p. 012034). IOP Publishing.

·         The introduction and related work sections should be extended to the more detailed background that would be supported by some literature. Add more recent reference to enhance literature survey section. Discuss the state-of-art techniques with their merits and issues. The literature should be developed and, if possible, presented in papers published in 2021 and 2022.

·         The "1.1. Monolithic Active Pixel Sensors", “Lassena family of MAPS” and “2. Previous results ”sections are the same as the article below. Also the references are very similar to this article.

o   Pritchard, J. L. , Velthuis, J. J., Beck, L., De Sio, C., & Hugtenburg, R. (2020). High resolution MLC leaf position measurements with a large area MAPS. IEEE Transactions on Radiation and Plasma Medical Sciences, 5(3), 392 - 397. https://doi.org/10.1109/TRPMS.2020.3007859

·         What solution you propose to make the system more robust. What is your difference from similar studies?

·         You should submit more experimental study results for your work. You should also provide comparisons with similar studies.

·         Rewrite the conclusion with following comments:
- Highlight your analysis and reflect only the important points for the whole paper.
- Mention the implication in the last of this section. Please, carefully review the manuscript to resolve these issues.

Author Response

Please see attached our replies

Reviewer 3 Report

I was asked to review the manuscript “Performance of a full-scale upstream MAPS-based verification device for radiotherapy”. It describes a Si imager for machine quality assurance in radation therapy. Generally, the quality of the development and the tests are good enough to justify a publication. However, I identified the following major shortcomings:

1)      The paper is too long. The only novelty concerns the extension of previous studies to a 2x2 stitched large-area imager. This can be reported if the manuscript is shortened by a factor of 2. This especially concerns the number of figures. I don’t think that 26 figures are necessary to convey the main message.

2)      Parts of the manuscript are not original. Figures 1-2 are the same as in Ref. 4. Similarly, the authors recite on several occasions the achievements of previous studies –even in the abstract.

Detailed comments:

·         L. 8: Much more is necessary to infer on the dose distribution in the patient. The sensor cannot measure the photon energy and good dose engines are necessary.

·         I think the language is a mixture of B.E. (e.g. “ionising”) and A.E. (“minimize”).

·         L. 33: I think the argument regarding the advantages of the upstream mounting is not correct. The upstream position only checks the machine, while an EPID allows to verify the interaction of the field with the patient. I think these options can only be compared on the level of a risk analysis (which is missing here).

·         L. 42: radiation hardness is mentioned as requirement. However, neither the reference nor the current study deal with radiation hardness.

·         Fig. 1: it should be clearer indicated where the “sensing cell” and the “active layer” are located.

·         L. 67: The authors tend to cite themselves. I think the claim of a pioneering work for RT verification requires a paper about the successful clinical use. Technical papers and patents are not sufficient.

·         L. 92: There was no MU definition provided. I think the dose rate is a better quantity.

·         L. 118/333: please explain “PCB”/”FFF”

·         Fig. 3: I think the sensor and the foam have to be indicated on the photo. The vertical U-shaped structure needs an explanation in the caption.

·         L. 133, ‘FFF’ should be explained

·         Fig. 4.: I think the (possible) angle should be indicated which is discussed in the text.

·         L. 154: please avoid “usual way”, “data are”?

·         L. 160: What “variations” do you refer to?

·         L. 169: “The angle is …”

·         L. 171: please indicate “trapezoidal” in Fig. 4 and discuss the consquences if they are relevant.

·         L. 219: I wonder about the relation of Eq. 1 to the auto-correlation. I think Eq. 1 is neither auto nor correlation. It’s more RMSE/chi-square.

·         L. 254: why was a new experiment necessary? In a 2x2 configuration there are always gaps in parallel/orthogonal to the leaves.

·         L. 268: ‘range’ with respect to?

·         L. 278: please explain how the ‘+-9 um’ are derived.

·         L. 292: “Here, it was shown …’ I think this was not the contents of the paper at all.

Author Response

Please see attached our replies.

Round 2

Reviewer 2 Report

The authors have made efforts to improve the quality of the article. Literature research is insufficient. The number of references presented in the article are very few.  The introduction and related work sections should be extended to the more detailed background that would be supported by some literature. Add more recent reference to enhance literature survey section. Discuss the state-of-art techniques with their merits and issues. The literature should be developed and, if possible, presented in papers published in 2021 and 2022.

Reviewer 3 Report

I was asked to review the revision of the manuscript “Performance of a full-scale upstream MAPS-based verification device for radiotherapy”. It describes a Si imager for machine quality assurance in radation therapy. The major issues identified in the initial version have been successfully addressed by the authors.

Please regard the following minor issues:

-          “MAPS” is not defined in the abstract.

-          P. 6., l. 177: please explain how the leaf edge is represented/evaluated for Fig. 7a

-          P. 7, l. 203: “newly developed” is clearly exaggerated. I think it’s a simple fit, which was carefully chosen to meet the requirements of the application.

-          Figures are sometimes hard to read on a print-out. Consider bold/bigger fonts and bigger marker sizes/strokes.

-          Still, the cited literature could be improved. The authors tend to cite themselves and I assume much more relevant studies on similar approaches are out there.

Round 3

Reviewer 2 Report

I have reviewed the revised manuscript title "Performance of a full-scale upstream MAPS-based verification device for radiotherapy". After revising my initial comments and comparing the changes, done by the authors, with them. I found that the authors addressed and answered most of the comments efficiently. Overall, the revised manuscrip is well organized and carefully prepared. The response letter was elegant and satisfactory. I thank the authors for their kind responses. The authors have sufficiently address my all comments. So, I think it is appropriate to accept the revised article. The authors have addressed all the concerns and responded to the review comments. The manuscript can be published in this journal.

Author Response

We woud like to thank the reviewer again for helping us shaping the manuscript.